# A Comprehensive Review of Small-Molecule Inhibitors Targeting Bruton Tyrosine Kinase: Synthetic Approaches and Clinical Applications

**DOI:** 10.3390/molecules28248037

**Published:** 2023-12-11

**Authors:** Qi Zhang, Changming Wen, Lijie Zhao, Yatao Wang

**Affiliations:** 1Nanyang Central Hospital, Nanyang 473000, China; zhangqizzu@126.com (Q.Z.); 13838729696@sina.com (C.W.); 2First People’s Hospital of Shangqiu, Shangqiu 476100, China; 3The Rogel Cancer Center, Department of Internal Medicine, University of Michigan, Ann Arbor, MI 48109, USA; 4Department of Orthopedics, China-Japan Union Hospital, Jilin University, Changchun 130033, China

**Keywords:** BTK, synthesis, application, small-molecule, drugs

## Abstract

Bruton tyrosine kinase (BTK) is an essential enzyme in the signaling pathway of the B-cell receptor (BCR) and is vital for the growth and activation of B-cells. Dysfunction of BTK has been linked to different types of B-cell cancers, autoimmune conditions, and inflammatory ailments. Therefore, focusing on BTK has become a hopeful approach in the field of therapeutics. Small-molecule inhibitors of BTK have been developed to selectively inhibit its activity and disrupt B-cell signaling pathways. These inhibitors bind to the active site of BTK and prevent its phosphorylation, leading to the inhibition of downstream signaling cascades. Regulatory authorities have granted approval to treat B-cell malignancies, such as chronic lymphocytic leukemia (CLL) and mantle cell lymphoma (MCL), with multiple small-molecule BTK inhibitors. This review offers a comprehensive analysis of the synthesis and clinical application of conventional small-molecule BTK inhibitors at various clinical stages, as well as presents promising prospects for the advancement of new small-molecule BTK inhibitors.

## 1. Introduction

BTK plays a crucial role as an essential enzyme in the signaling pathway of BCR activation [1,2,3]. B-cells, a type of white blood cell, play a crucial role in producing antibodies and depend on them for their growth and functioning. The abnormal activity of BTK has been linked to different B-cell cancers, autoimmune conditions, and inflammatory disorders [4,5].

At present, the utilization of small-molecule inhibitors targeting BTK has become a very promising treatment method for these diseases. These inhibitors work by binding to the active site of BTK and blocking its enzymatic activity, thereby inhibiting downstream signaling pathways that promote cell proliferation, survival, and inflammation [6]. Ibrutinib, the initial BTK inhibitor approved by the U.S. Food and Drug Administration (FDA), has demonstrated exceptional effectiveness in addressing B-cell malignancies like CLL and MCL [7]. However, Ibrutinib, the first-generation BTK inhibitor, still possesses certain limitations. It possesses the ability to inhibit various targets, including epidermal growth factor receptor (EGFR), tyrosine kinase expressed in hepatocellular carcinoma (TEC), bone marrow kinase on chromosome X (BMX), and others [8]. In order to solve these difficulties, many enterprises have begun to develop the second generation of BTK inhibitors [9,10]. Approved in 2017 and 2019, respectively, Acalabrutinib and Zanubrutinib have been structurally optimized and improved to have less off-target effects and greater effectiveness than Ibrutinib [11]. Since then, several other small-molecule BTK inhibitors have been created and are currently undergoing assessment in clinical trials for various indications. In the last ten years, a multitude of preclinical and clinical trials have examined the efficacy of BTK inhibitors as standalone treatments or in conjunction with targeted therapies, immunotherapy, or traditional chemotherapy to treat different types of cancers. In addition, as covalent BTK inhibitors still have off-target effects and tolerability problems, the industry and academia are increasingly calling for non-covalent BTK inhibitors. A new generation of BTK inhibitors, such as Pirtobrutinib, is in the fast lane of development [12,13].

The progress in the realm of BTK inhibitor research heralds a promising spectrum of therapeutic avenues for the effective management of afflictions characterized by BTK overexpression. Additional investigation and inquiry are essential to achieving a comprehensive understanding of the full potential inherent in these inhibitors. Further scrutiny and inquiry are paramount to attaining a comprehensive grasp of the inherent potential embodied by these inhibitors. As far as our current knowledge extends, a thorough investigation into the synthetic methodologies employed in chemical compounds and their respective mechanisms of action within clinical contexts holds substantial promise for propelling the advancement of groundbreaking pharmaceuticals.

Drawing upon well-substantiated data concerning BTK inhibitors, this review methodically elucidates the clinical utility and synthetic techniques associated with prototypical BTK inhibitors across diverse clinical phases (Appendix A). This information carries significant import in the ongoing design and refinement of BTK inhibitors.

## 2. Signaling Pathway of BTK

BTK is a non-receptor tyrosine kinase that plays a crucial role in the signaling pathways of B-cells, macrophages, and microglia [14]. As shown in Figure 1, BTK is activated downstream of the BCR and FcγRIII in macrophages and microglia, leading to the activation of downstream elements crucial to immune cell function. When BTK is activated, it phosphorylates phospholipase Cγ2 (PLCγ2), causing the hydrolysis of phosphatidylinositol 4,5-bisphosphate (PIP2) to produce inositol 1,4,5-trisphosphate (IP3) and diacylglycerol (DAG). IP3 binds to its receptor on the endoplasmic reticulum, leading to the release of calcium ions into the cytoplasm, which activates downstream signaling pathways. DAG triggers the activation of protein kinase C (PKC), which subsequently initiates the activation of signaling pathways downstream. BTK also activates the nuclear factor kappa B (NF-κB) pathway, which is involved in the regulation of immune responses. BTK also participates in signaling networks that play a role in innate immunity, such as the transmission of signals through the Fc epsilon receptor (FcεR) in mast cells and basophils. When antigen binds to immunoglobulin E (IgE) molecules linked to FcεR, it triggers the phosphorylation of immunoreceptor tyrosine-based activation motifs (ITAMs) in the FcεR β and γ chains. During the BCR signal transduction process, the phosphorylation of FcεR and ITAMs plays a crucial role in recruiting and subsequently activating Lck/yes-related novel protein tyrosine kinase (LYN) and spleen tyrosine kinase (SYK).

## 3. Representative Small-Molecule BTK Inhibitors in the Clinic

### 3.1. Aminopyrimidines

#### 3.1.1. Spebrutinib (CC-292)

Spebrutinib, developed by Xinji Company, is a small-molecule drug currently in clinical phase II. It is being investigated for its potential in treating diffuse large B-cell lymphoma (DLBCL) and CLL. Spebrutinib irreversibly binds to the cysteine residue (Cys481) in the active site of BTK, inhibiting its activity and downstream signaling pathways (IC_50_ = 0.5 nM) [15,16]. In clinical trials, Spebrutinib has shown favorable efficacy in patients with relapsed or refractory CLL and MCL [17]. It has demonstrated durable responses and manageable toxicity profiles, leading to its potential as a targeted therapy for these diseases. However, additional clinical trials are required to determine its effectiveness and safety in larger groups of patients [17].

In 2008, a method for synthesizing Spebrutinib was elucidated (Figure 1) [18,19]. First, the 4-Cl of 2,4-dichloro-5-fluoropyrimidine (**SPEB-001**) is substituted with *tert*-butyl (3-aminophenyl)carbamate (**SPEB-002**) in the presence of *N*,*N*-diisopropylethylamine (DIPEA) to obtain **SPEB-003**. **SPEB-003** undergoes further substitution with 4-(2-methoxyethoxy)aniline (**SPEB-004**) to obtain **SPEB-005**. Treatment of **SPEB-005** with trifluoroacetic acid (TFA) to remove the *t*-butyloxycarbonyl (Boc) group gives **SPEB-006**. Finally, in the presence of DIPEA, **SPEB-006** is amidated with acryloyl chloride to obtain Spebrutinib.

#### 3.1.2. Evobrutinib (M-2951)

Evobrutinib, developed by Merck Serrano, is a small-molecule medication currently in the advanced clinical phase III of development. It is specifically designed to address the treatment of multiple sclerosis (MS). In clinical trials, Evobrutinib has shown efficacy in treating relapsing forms of MS. Phase 2 trials showed a notable decrease in the occurrence of fresh brain lesions and relapse rates when compared to the placebo group [20,21]. Additionally, Evobrutinib has shown potential to treat autoimmune diseases such as rheumatoid arthritis (RA) and systemic lupus erythematosus (SLE) [22]. Although additional research is required to comprehensively comprehend its long-term safety profile, Evobrutinib demonstrates promise as a valuable therapeutic choice for individuals suffering from autoimmune disorders.

The preparation of Evobrutinib first consists of the reaction of 5,6-dichloropyrimidin-4-amine (**EVOB-001**) with *N*-Boc piperidine **EVOB-002** to give **EVOB-003** (Figure 2) [23]. Subsequently, Suzuki coupling of **EVOB-003** with (4-phenoxyphenyl)boronic acid (**EVOB-004**) catalyzed by Pd(OAc)_2_ affords **EVOB-005**. Treatment of **EVOB-005** with HCl to remove the Boc moiety gives **EVOB-006**. Finally, **EVOB-006** is amidated with acryloyl chloride to obtain Evobrutinib.

#### 3.1.3. Remibrutinib (LOU-064)

Remibrutinib is a small-molecule drug developed by Novartis pharmaceuticals. At present, the highest research and development stage of the drug is clinical phase III, which is used to treat chronic urticaria (CU), MS, urticaria, and relapsing remitting MS. Remibrutinib is a highly effective BTK inhibitor that can be taken orally. It has a remarkable potency, with an IC_50_ value of 1 nM. In blood, Remibrutinib demonstrates strong inhibition of BTK activity, with an IC_50_ value of 0.023 μM [24]. Remibrutinib effectively suppressed HuMOG experimental autoimmune encephalomyelitis (EAE), a B-cell-dependent disease, in a dose-dependent manner and significantly alleviated neurological symptoms. When administered orally at a dose of 30 mg/kg, Remibrutinib exhibited potent inhibition of BTK in both peripheral immune organs and the brain of EAE mice. In clinical trials, Remibrutinib has shown promising efficacy in patients with CLL and other B-cell malignancies. It has shown impressive response rates and long-lasting remissions in patients who have not responded to previous treatments. Additionally, Remibrutinib has shown a favorable safety profile with manageable toxicities, including mild to moderate gastrointestinal symptoms and reversible hematological abnormalities [25,26].

Figure 3 presents a concise method for synthesizing Remibrutinib [24]. First, in the presence of ammonia, 4,6-dichloro-5-methoxypyrimidine (**REMI-001**) undergoes halogen amination to give the amine **REMI-002**, and then boron tribromide is used to remove the methyl group of **REMI-002** to give the phenol **REMI-003**. In the presence of diisopropyl azodicarboxylate (DIAD) with Smopex-301, a convenient polymer-supported version of triphenylphosphine, **REMI-003** reacts with hydroxyethylamine **REMI-004** in a Mitsunobu reaction to give **REMI-005**. Under the catalytic effect of PdCl_2_(PPh_3_)_2_, **REMI-005** reacts with the intermediate **REMI-006** in a Suzuki coupling reaction to give **REMI-007**. The Boc group of **REMI-007** is removed using TFA to obtain **REMI-008**. Finally, **REMI-008** is amidated with acrylic acid to obtain Remibrutinib in the presence of the condenser propylphosphonic anhydride (T3P).

### 3.2. Pyrimidine-Fused Bicyclic Heterocycles

#### 3.2.1. Ibrutinib (Imbruvica, PCI-32765)

Ibrutinib, originally created by Pharmacyclics LLC. and marketed as Imbruvica, received its first FDA approval to treat MCL on 13 November 2013. Subsequently, it obtained FDA approval for the management of Waldenstrom macroglobulinemia (WM), CLL, marginal zone B-cell lymphoma (MZBL), and graft vs. host disease (GvHD) in subsequent years. Ibrutinib is a BTK inhibitor that is both selective and irreversible, with an IC_50_ value of 0.5 nM [27]. Ibrutinib effectively blocks the activity of the enzyme by creating a covalent bond with Cys481, found in the active site of BTK. This blocking action of Ibrutinib on BTK then stops the phosphorylation process of downstream substrates like phospholipase C-γ (PLC-γ) [28]. The increase in IC_50_ against BTK-C481S phosphorylation from 2.2 nM to 1 μM is due to the fact that Ibrutinib cannot establish a covalent bond with the serine hydroxyl group [29].

An optimized synthetic route for Ibrutinib is described in Figure 4 [30]. First, 4-phenoxybenzoyl chloride (**IBRU-001**) undergoes a Knoevenagel-type reaction with malony-dinitril to obtain the enol intermediate, followed by the methylation of the enol intermediate with dimethyl sulfate to obtain **IBRU-002**. In the presence of triethylamine (TEA), **IBRU-002** undergoes pyrazole cyclization with hydrazine **IBRU-003** to give **IBRU-004**. Finally, catalyzed by acetic acid, **IBRU-004** undergoes a pyrimidine cyclization with the cycloadduct dimethylformamide dimethylacetal (DMF-DMA) to obtain Ibrutinib.

#### 3.2.2. Zanubrutinib (Brukinsa, BGB-3111)

Zanubrutinib, developed by BeiGene, received its initial FDA approval on 14 November 2019, to treat MCL. Subsequently, it gained approval to treat MZBL, CLL, and WM. Zanubrutinib, a member of the second-generation BTK inhibitor drug category, demonstrates exceptional effectiveness and selectivity towards BTK while causing minimal off-target effects in comparison [31]. Due to its similar binding specificity to other BTK inhibitors, Zanubrutinib impedes BTK function by forming a covalent bond with Cys481. Due to this binding pattern, Zanubrutinib has the ability to bind to adenosine triphosphate (ATP)-binding kinases that possess a Cys481 at this specific site, regardless of their relationship or similarity, exhibiting varying levels of affinity [31,32,33].

Zanubrutinib is prepared by the reaction of 4-phenoxybenzoic acid (**ZANU-001**) with thionyl chloride to obtain the chloride intermediate **IBRU-001**, and then **IBRU-001** undergoes a Knoevenagel-type reaction with malony-dinitril to obtain **ZANU-002** in the presence of DIPEA (Figure 5) [34]. Alkylation of **ZANU-002** with trimethyl orthoformate affords **IBRU-002**, followed by cyclization of **IBRU-002** with hydrazine hydrate to give pyrazol **ZANU-003**. Under the catalytic effect of acetic acid, **ZANU-003** reacts with *N*-Boc piperidine **ZANU-004** to give **ZANU-005**. Reduction of **ZANU-005** using sodium borohydride gives **ZANU-006**, followed by hydrolysis of the cyano group of **ZANU-006** using hydrogen peroxide to give the amide **ZANU-007**. Treatment of **ZANU-007** with TFA to remove the Boc group yields the trifluoroacetate salt **ZANU-008**. Finally, **ZANU-008** is first amidated with acryloyl chloride to obtain the amide **ZANU-009**, and then **ZANU-009** is separated using Chiral pre-high-performance liquid chromatography (HPLC) to obtain Zanubrutinib.

#### 3.2.3. Tirabrutinib (Velexbru, ONO/GS-4059)

Tirabrutinib Hydrochloride, created by Ono Pharmaceutical, was granted approval by the Pharmaceuticals and Medical Devices Agency (PMDA) on 25 March 2020. This drug is used for treating lymphoma [35]. Tirabrutinib, an orally administered BTK inhibitor (IC_50_ = 6.8 nM), possesses the capability to cross the blood–brain barrier (BBB) [36]. As a novel BTK inhibitor, Tirabrutinib exerts its anti-tumor and anti-inflammatory effects through selective inhibition of BTK activity. For example, Tirabrutinib has shown good efficacy in patients with CLL [37,38]. Common adverse events (AEs) were rash (35.3%) and vomiting (29.4%) [38].

Tirabrutinib Hydrochloride is prepared first by the reaction of dibenzylamine with nitropyrimidine **TIRA-001** to give **TIRA-002** (Figure 6) [39]. In the presence of TEA, the reaction of **TIRA-002** with aminopyrrolidine **TIRA-003** gives amine **TIRA-004**. Reduction of **TIRA-004** using zinc powder affords the amine **TIRA-005**, which is subsequently amidated with *N*,*N*′-carbonyl diimidazole (CDI) to give imidazolidinone **TIRA-006**. Under the 20% Pd(OH)_2_/C reduction, **TIRA-007** is derived from **TIRA-006** by removing the benzyl (Bn) group. A Chan–Lam coupling reaction of amide **TIRA-007** with boronic acid **TIRA-008** catalyzed by copper acetate gives **TIRA-009**, which is treated with hydrochloric acid to eliminate the Boc-protecting group, resulting in the formation of **TIRA-010**. In the presence of 1-(3-dimethylaminopropyl)-3-ethylcarbodiimide hydrochloride (EDC) and hydroxybenzotriazole (HOBt), **TIRA-010** is amidated with 2-butynoic acid (**TIRA-011**) to obtain **TIRA-012**. Finally, treatment of **TIRA-012** with hydrochloric acid gives Tirabrutinib Hydrochloride.

#### 3.2.4. Nemtabrutinib (MK-1026)

Nemtabrutinib was developed by Arqule Inc. At present, the highest research and development stage of the drug is clinical phase III, which is used to treat chronic lymphoblastic leukemia. Nemtabrutinib is a promising BTK inhibitor with potent activity against B-cell malignancies, including CLL and non-Hodgkin lymphoma (NHL) [40]. It has also shown promising activity against drug-resistant forms of these cancers. In clinical trials, Nemtabrutinib has shown favorable efficacy and safety profiles [41]. Nemtabrutinib has exhibited substantial clinical efficacy in patients with relapsed or refractory CLL and NHL, resulting in long-lasting responses. Moreover, it has displayed promise when used in combination with other targeted agents, thereby augmenting its effectiveness [42].

The approach for Nemtabrutinib is described in Figure 7 [43]. First, pyrrolo[2,3-d]pyrimidine **NEMT-001** undergoes lithium-bromine exchange with a lithium reagent to obtain an organolithium reagent, which is then added to methyl 2-chloro-4-phenoxybenzoate (**NEMT-002**) to obtain ketone **NEMT-003**. Finally, in the presence of DIPEA, the Cl of **NEMT-003** is substituted with the amine **NEMT-004** to obtain Nemtabrutinib.

#### 3.2.5. Rilzabrutinib (PRN-1008)

Rilzabrutinib, created by Principia Biopharma Inc., is currently in the advanced clinical phase III of development. Its primary purpose is to treat idiopathic thrombocytopenic purpura (ITP). Rilzabrutinib, an orally active inhibitor of BTK (IC_50_ = 1.3 nM), is a reversible, covalent, and selective compound [44,45]. Rilzabrutinib has demonstrated encouraging effectiveness in the management of autoimmune disorders during clinical trials [46]. In a phase II trial involving patients with active RA, Rilzabrutinib significantly improved disease activity compared to placebo. Additionally, Rilzabrutinib has shown efficacy in the treatment of ITP and pemphigus vulgaris, with positive results observed in phase 2 trials [47].

The preparation of Rilzabrutinib is depicted in Figure 8 [48]. First, the amine **RILZ-001** is iodinated using *N*-iodosuccinimide (NIS) to obtain **RILZ-002**. In the presence of triphenylphosphine and DIAD, **RILZ-002** is mixed with *tert*-butyl (*R*)-3-hydroxypiperidine-1-carboxylate (**RILZ-003**) in a Mitsunobu reaction to give **RILZ-004**. Catalyzed by Pd(dppf)Cl_2_, **RILZ-004** undergoes a Suzuki coupling reaction with the intermediate **RILZ-005** to give **RILZ-006**. **RILZ-006** is treated with TFA to remove the Boc group to give **RILZ-007**. Subsequently, **RILZ-007** is amidated with 2-cyanoacetic acid in the presence of the condenser 2-(7-azabenzotriazol-1-yl)-*N*,*N*,*N*′,*N*′-tetramethyluronium hexafluorophosphate (HATU) to give **RILZ-008**. Finally, **RILZ-008** undergoes a Knoevenagel condensation reaction with the aldehyde **RILZ-009** to obtain Rilzabrutinib.

#### 3.2.6. Abivertinib (AC-0010)

Abivertinib, developed by Zhejiang ACEA Pharmaceutical Co., Ltd., has applied for listing to treat non-small cell lung cancer (NSCLC) in China. Abivertinib is a specific inhibitor of tyrosine kinases that targets mutant forms of human EGFR and BTK [49]. Abivertinib has been investigated for its potential use in the management of NSCLC and B-cell malignancies. It exerts its effects by binding to and inhibiting EGFR and BTK receptors, leading to immunomodulatory actions. These actions include the inhibition of pro-inflammatory cytokine production and release, such as tumor necrosis factor (TNF)-α and interleukins [50]. It has also shown favorable pharmacokinetic properties, with good oral bioavailability and tissue distribution. Nevertheless, additional research is required to thoroughly assess its long-term effectiveness and safety.

The preparation of Abivertinib starts with the *N*-alkylation of pyrrolo[2,3-d]pyrimidine **ABIV-001** with 2-(trimethylsilyl)ethoxymethyl chloride (SEMCl) in the presence of sodium hydride to yield **ABIV-002** (Figure 9) [51]. Subsequently, in the presence of potassium carbonate, **ABIV-002** undergoes a Williamson synthesis with 3-nitrophenol (**ABIV-003**) to give the ether **ABIV-004**. The Buchwald–Hartwig cross-coupling of **ABIV-004** with aniline **ABIV-005** catalyzed by Pd_2_dba_3_ affords **ABIV-006**, which is reduced with iron powder to give the amine **ABIV-007**. Under the action of DIPEA, **ABIV-007** is amidated with acryloyl chloride to obtain **ABIV-008**. Finally, **ABIV-008** is subjected to incomplete removal of the SEM group by TFA to obtain the hydroxymethyl intermediate **ABIV-009**, and then ammonia treatment is applied to remove the hydroxymethyl group to obtain Abivertinib.

### 3.3. Benzopyrroles

#### 3.3.1. Luxeptinib (CG-806)

Luxeptinib, co-developed by Crystalgenomics Inc. and Aptose, is investigated for its potential in treating various conditions, including myelodysplastic syndrome (MDS), CLL, acute myeloid leukemia (AML), and NHL in clinical phase I. Luxeptinib is a non-covalent, reversible, and orally active inhibitor that targets both fms-like tyrosine kinase (FLT3) and BTK [52]. Luxeptinib effectively blocks the phosphorylation of AKT, ERK1/2, Plcg2, BTK, and S6 ribosomal proteins and significantly suppresses SYK phosphorylation in primary CLL cells [53]. This implies that Luxeptinib may be applicable to patients with different types of B-cell malignancies who have developed resistance, refractory response, or intolerance to covalent or non-covalent BTK inhibitors.

One notable method of Luxeptinib is depicted in Figure 10 [54,55,56]. First, 7-bromo-3-oxoisoindoline-4-carbonitrile (**LUXE-001**) is reduced using Raney Ni to give the aldehyde **LUXE-002**. In ammonia, **LUXE-002** is condensed with 2-oxopropanal (**LUXE-003**) to give the imidazole **LUXE-004**. The Suzuki coupling of **LUXE-004** with the borate ester **LUXE-005** catalyzed by Pd(PPh_3_)_4_ yields **LUXE-006**. Finally, 2,4,6-trifluorobenzoyl azide (**LUXE-007**) undergoes a Curtius rearrangement to give the isocyanate, which reacts with the amine **LUXE-006** to give the urea Luxeptinib.

#### 3.3.2. Branebrutinib (BMS-986195)

Branebrutinib, created by Bristol Myers Squibb, is currently in phase II development. This drug shows promise in treating various conditions, including atopic dermatitis (AD), Sjogren syndrome (SS), RA, and SLE. In preclinical studies, Branebrutinib has demonstrated potent inhibition of BTK activity (IC_50_ = 0.1 nM), leading to the suppression of B-cell activation and a subsequent reduction in disease severity [57]. The results indicate that Branebrutinib shows promise for treating autoimmune disorders such as RA and SLE [58]. Promising outcomes have been observed in clinical trials assessing the effectiveness of Branebrutinib. In a phase II trial involving individuals with relapsed or refractory MCL, Branebrutinib exhibited a high overall response rate (ORR) and long-lasting responses [59]. Additionally, in a phase III trial for RA, Branebrutinib showed significant improvement in disease activity compared to placebo.

The synthesis of Branebrutinib begins with a Pd_2_(dba)_3_-catalyzed Buchwald–Hartwig cross-coupling reaction of indole **BRAN-001** with *tert*-butyl (*S*)-piperidin-3-ylcarbamate (**BRAN-002**) to give **BRAN-003** (Figure 11) [57]. Subsequently, **BRAN-003** undergoes the hydrolysis of cyanide in the presence of H_2_SO_4_ to give the amide **BRAN-004**. Finally, in the presence of the condenser HATU, **BRAN-004** undergoes amidation with 2-butynoic acid (**TIRA-011**) to obtain Branebrutinib.

#### 3.3.3. Elsubrutinib (ABBV-105)

Elsubrutinib, developed by Abbvie Ltd., is in clinical phase II and is used to treat SLE and RA. Elsubrutinib is a BTK inhibitor that is taken orally and exhibits selectivity and irreversibility (IC_50_ = 0.18 μM) [60]. Elsubrutinib irreversibly inhibits BTK and exhibits superior selectivity in the kinome. It demonstrates potent activity in cellular assays involving B-cell receptors, Fc receptors, and TLR-9. Its mode of action involves the irreversible inhibition of BTK, which disrupts BCR signaling. Both preclinical and clinical research have confirmed its effectiveness and tolerable toxicity profile, positioning it as a promising treatment option for individuals with B-cell malignancies and autoimmune diseases [61]. Elsubrutinib inhibits antibody responses to both non-dependent thymus and dependent thymus antigens, reducing paw swelling and bone destruction in collagen-induced arthritis in rats. It also alleviates disease in an IFNα-accelerated lupus nephritis model [60].

The synthesis of Elsubrutinib starts with the Suzuki coupling of 4-bromo-1*H*-indole-7-carboxamide (**ELSU-001**) with the borate ester **ELSU-002** catalyzed by Pd(dppf)Cl_2_ to obtain **ELSU-003** (Figure 12) [62]. **ELSU-004** is obtained by catalytic hydrogenation of **ELSU-003** with H_2_ and Pd/C. Subsequently, the Boc group is removed using acetyl chloride to obtain **ELSU-005**, which is amidated with acryloyl chloride in the presence of DIPEA to obtain **ELSU-006**. Finally, **ELSU-006** is separated into the desired isomer of Elsubrutinib by HPLC.

### 3.4. Pyrazine-Fused Bicyclic Heterocycles

#### 3.4.1. Acalabrutinib (Calquence, ACP-196)

On 31 October 2017, the FDA authorized the initial approval of Acalabrutinib to treat MCL. Subsequently, it obtained FDA approval for the treatment of CLL as well on 3 August 2022. Calquence, originally developed by AstraZeneca and Acerta Pharma LLC., is the brand name for Acalabrutinib. Acalabrutinib is a second-generation BTK inhibitor that is taken orally. It is irreversible and highly selective [63]. Acalabrutinib effectively inhibits BTK with a potency of 30 nM (IC_50_) and effectively suppresses CD69 B-cell activation in human whole blood with an EC_50_ of 8 nM. It works by binding to a Cys481 residue in the active site of BTK, forming a covalent bond, and inhibiting BTK activity [64]. Because Acalabrutinib can effectively block the activation of CD86 and CD69 downstream signaling proteins through BTK, it effectively suppresses the growth and survival of malignant B-cells. Acalabrutinib demonstrates improved selectivity and inhibition of BTK activity compared to Ibrutinib. Additionally, it exhibits significantly higher IC_50_ values or minimal inhibition of other kinase activities, including EGFR, ERBB4, tyrosine-protein kinase (ITK), Janus kinase 3 (JAK3), hematopoietic cell kinase (HCK), B lymphoid tyrosine kinase (BLK), feline Gardner-Rasheed sarcoma viral oncogene homolog (FGR), and SRC [65].

A method of synthesis for Acalabrutinib is summarized in Figure 13 [66]. First, 3-chloropyrazine-2-carbonitrile (**ACAL-001**) is catalytically reduced by hydrogen to give the amine **ACAL-002**. In the presence of the condenser HATU, **ACAL-002** is amidated with ((benzyloxy)carbonyl)-L-proline (**ACAL-003**) to give the amide **ACAL-004**. The cyclization of **ACAL-004** in acetonitrile using phosphorus oxychloride with 1,3-dimethyl-2-imidazolidinone affords the imidazole **ACAL-005**. Subsequently, **ACAL-005** is brominated using *N*-bromosuccinimide (NBS) to obtain **ACAL-006**. **ACAL-006** undergoes a halogen aminolysis reaction in ammonia to give **ACAL-007**. Under the catalysis of Pd(dppf)Cl_2_, **ACAL-007** undergoes a Suzuki coupling reaction with (4-(pyridin-2-ylcarbamoyl)phenyl)boronic acid (**ACAL-008**) to give **ACAL-009**. Treatment of **ACAL-009** with 33% HBr to remove the benzyloxycarbonyl (Cbz) group gives **ACAL-010**. Finally, **ACAL-010** is amidated with 2-butynoic acid (**TIRA-011**) in the presence of the condenser HATU to obtain Acalabrutinib.

#### 3.4.2. Fenebrutinib (GDC-0853)

Fenebrutinib, a small-molecule drug, was developed by Genentech. At present, the highest research and development stage of the drug is clinical phase III, which is used to treat MS and chronic progressive MS. Fenebrutinib is a highly effective and specific non-covalent BTK inhibitor that can be taken orally. The inhibitor has Kis values of 0.91 nM, 1.6 nM, 1.3 nM, 12.6 nM, and 3.4 nM for wild-type BTK, as well as the C481S, C481R, T474I, and T474M mutants [67]. It has also shown synergy with other targeted therapies, such as Venetoclax, further enhancing its anti-tumor effects. In clinical trials, Fenebrutinib has shown promising efficacy in patients with CLL and other B-cell malignancies [68,69]. Fenebrutinib has exhibited impressive response rates and long-lasting remissions, both when used alone and in conjunction with other medications. Moreover, it has displayed promising results in patients with relapsed or refractory conditions who have not responded to prior treatments [70]. While Fenebrutinib had an acceptable safety profile, the primary end point, the SRI-4 response, was not met despite the evidence of strong pathway inhibition [68].

The preparation of Fenebrutinib is depicted in Figure 14 [71]. First, 5-bromo-2-nitropyridine (**FENE-001**) undergoes Buchwald–Hartwig cross-coupling with (*S*)-*tert*-butyl 3-methylpiperazine-1-carboxylate (**FENE-002**) catalyzed by Pd_2_(dba)_3_ to give **FENE-003**. The amine **FENE-004** is obtained by reducing **FENE-003**. A Buchwald–Hartwig cross-coupling reaction of **FENE-004** with pyridinone **FENE-005** catalyzed by Pd_2_(dba)_3_ affords **FENE-006**. Treatment of **FENE-006** with HCl to remove the Boc moiety yields **FENE-007**. A reductive amination of **FENE-007** with oxetan-3-one (**FENE-008**) under the reducing action of sodium cyanoborohydride yields **FENE-009**. Subsequently, **FENE-009** undergoes a Miyaura Borylation reaction with bis(pinacolato)diboron catalyzed by Pd_2_(dba)_3_ to obtain the borate ester **FENE-010**. Then, **FENE-010** undergoes a Suzuki coupling reaction with the intermediate **FENE-011** to obtain **FENE-012**. Finally, the aldehyde group of **FENE-012** is reduced using sodium borohydride to obtain Fenebrutinib.

### 3.5. Others

#### 3.5.1. Orelabrutinib (ICP-022)

Orelabrutinib, created by Beijing InnoCare Pharma Tech, received approval from the National Medical Products Administration (NMPA) on 25 December 2020, for its effectiveness in treating MCL and CLL [72]. The NMPA also approved the drug to treat MZBL on 20 April 2023. Orelabrutinib, a potent oral inhibitor of BTK, exhibits remarkable efficacy as an antineoplastic agent. By blocking the activation of the B-cell antigen receptor signaling pathway and subsequent survival pathways triggered by BTK, it effectively inhibits the growth of cancerous B-cells with elevated levels of BTK [73]. In terms of safety, the AEs observed during the study of Orelabrutinib treatment are generally consistent with the characteristics of BTK inhibitors. The most common AEs are hematologic toxicities, including thrombocytopenia, neutropenia, and anemia. In addition, a small number of patients have also reported respiratory system infections and purpura [74].

The preparation of Orelabrutinib begins with the hydrolysis of 2,6-dichloronicotinonitrile (**OREL-001**) in the presence of concentrated sulfuric acid to give the amide **OREL-002** (Figure 15) [75]. Subsequently, a Suzuki coupling reaction of **OREL-002** with (4-phenoxyphenyl)boronic acid (**OREL-003**) is catalyzed by Pd(dppf)Cl_2_ to obtain **OREL-004**, which undergoes further Suzuki coupling reaction with *N*-Boc-protected boronic ester **OREL-005** to obtain **OREL-006**. Catalytic hydrogenation of **OREL-006** by Pd/C gives **OREL-007**, which is treated with TFA to remove the Boc group. Finally, **OREL-008** is *N*-alkylated with acryloyl chloride in the presence of TEA to obtain Orelabrutinib.

#### 3.5.2. Pirtobrutinib (Jaypirca, LOXO-305)

Pirtobrutinib, developed by Redx Pharma, received FDA approval on 27 January 2023 to treat MCL. It is a highly selective and non-covalent-binding BTK inhibitor and demonstrates significant efficacy in suppressing different BTK-C481 substitution mutations. Pirtobrutinib exhibits a high level of selectivity for BTK, with a selectivity ratio of over 300-fold compared to the 370 other kinases tested. Additionally, at a concentration of 1 μM, it does not significantly impede any non-kinase off-targets [76,77]. Cys481 mutations play a significant role in conferring resistance to covalent BTK inhibitors, but they do not affect the efficacy of Pirtobrutinib. Although the precise mechanisms behind resistance to covalent BTK inhibitors are not yet fully understood, it is evident that these mutations play a major contributing factor [77,78,79].

The synthesis of Pirtobrutinib starts with the reaction of 5-fluoro-2-methoxybenzoic acid (**PIRT-001**) with thionyl chloride to obtain the chloride intermediate, and then the amidation of the chloride intermediate with 4-(aminomethyl)benzoic acid (**PIRT-002**) under the action of TEA to obtain amide **PIRT-003** takes place (Figure 16) [80]. Under the same conditions, **PIRT-003** is reacted with sulfoxide to obtain the chloride intermediate, and then **PIRT-004** is obtained by Knoevenagel condensation of the chloride intermediate with malony-dinitril. Alkylation of **PIRT-004** with trimethyl orthoformate affords the ether **PIRT-005**. Subsequently, hydrazine **PIRT-006** is transformed into the free base in TEA, followed by cyclization with **PIRT-005** in the presence of TEA to give pyrazole **PIRT-007**. Finally, **PIRT-007** is hydrolyzed in methanesulfonic acid (MsOH) to give Pirtobrutinib.

#### 3.5.3. Tolebrutinib (SAR-442168)

Tolebrutinib, created by Principia Biopharma Inc., is currently in the advanced clinical phase III of development. This drug is specifically designed to address MS and chronic progressive MS. Tolebrutinib is a highly effective and specific inhibitor of BTK, which can be taken orally and easily crosses the BBB. In Ramos B-cells, it has shown IC_50_ values of 0.4 nM, while in HMC microglia cells, it has exhibited IC_50_ values of 0.7 nM [81]. The mechanism of action of Tolebrutinib involves blocking the activation of BTK, thereby preventing the proliferation and survival of malignant B-cells. This inhibition also leads to the suppression of downstream signaling pathways, like NF-κB and AKT, which are crucial for the growth and survival of cancer cells [82]. In clinical trials, Tolebrutinib has shown favorable efficacy and safety profiles [83]. It has demonstrated significant clinical responses in patients with relapsed or refractory CLL and MCL, leading to its accelerated approval by the FDA for these indications [84]. The drug has also shown potential in other B-cell malignancies, such as WM and DLBCL [85].

The synthetic route of Tolebrutinib as described in the publication is shown in Figure 17 [86]. First, 2,4-dichloro-3-nitropyridine (**TOLE-001**) is reacted with *tert*-butyl (*R*)-3-aminopiperidine-1-carboxylate (**TOLE-002**) in the presence of TEA to give **TOLE-003**. Subsequently, **TOLE-003** is reacted with bis(4-methoxybenzyl)amine (**TOLE-004**) in the presence of TEA to give **TOLE-005**, which is reduced by iron powder to give aminopyridine **TOLE-006**. **TOLE-006** is cyclized with CDI to give **TOLE-007**, and treatment of **TOLE-007** with TFA to remove the Boc and *p*-methoxybenzyl (PMB) groups gives **TOLE-008**. The reaction of **TOLE-008** with (Boc)_2_O utilizes the Boc group to protect the piperidine ring to obtain **TOLE-009**, and the subsequent condensation of **TOLE-009** with DMF-DMA to obtain **TOLE-010** is performed. A Chan–Lam coupling reaction of **TOLE-010** with (4-phenoxyphenyl)boronic acid (**TOLE-011**) catalyzed by copper acetate yields **TOLE-012**. **TOLE-012** is stripped of its Boc group using hydrochloric acid to give **TOLE-013**. Finally, **TOLE-013** is amidated with acryloyl chloride to obtain Tolebrutinib.

## 4. Challenge and Prospective

At present, the difficulties in the advancement of BTK inhibitors mainly include: (1) off-target effects and related adverse reactions and (2) drug resistance. BTK belongs to the Tec family of non-receptor tyrosine kinases and is essential for BCR signaling. Nonetheless, BTK is found in different cell types, such as macrophages and mast cells, which may lead to unintended consequences when employing small-molecule inhibitors. Achieving selectivity for BTK while avoiding inhibition of other kinases is a significant challenge in the development of these inhibitors. After iteration and optimization, the current BTK inhibitors have improved significantly in terms of off-target adverse reactions in general, but another difficulty is the problem of drug resistance, which has not been addressed in the existing commercially available BTK inhibitors. Among them, acquired resistance caused by point mutations of BTK kinase is a major cause of resistance to BTK inhibitors; moreover, covalent binding site mutations (C481S, C481F/y, and C481R), “gated region” mutations (T474I and T474S), and β-fold VII mutations (L528W) were the major ones.

In response to these problems, the pharmacochemical strategies of the new generation of BTK inhibitors focus on optimizing existing molecules, such as developing non-covalent inhibitors, avoiding steric hindrance of mutant residues, interacting with mutant residues, modifying solvent-accessible regions, and developing new skeletons. In addition, strategies to combat resistance to BTK inhibitors include combination with other targeted drugs, reduction of BTK content, inhibition of upstream and downstream pathways of BTK, combination with CAR-T cell immunotherapy, and implementation of other pathways to inhibit the proliferation of tumor cells.

## 5. Conclusions

In conclusion, the use of small-molecule inhibitors targeting BTK has demonstrated significant potential for treating different types of B-cell malignancies and autoimmune disorders. These inhibitors effectively hinder the function of BTK, a key player in BCR signaling and immune response. Clinical trials have demonstrated their efficacy in improving patient outcomes, including ORR and progression-free survival (PFS). Furthermore, these inhibitors have shown favorable safety profiles with manageable adverse effects. The emergence of small-molecule BTK inhibitors signifies notable progress in targeted therapy and holds immense promise for the prospective management of BTK-related diseases.

## Data Availability

All of the relevant data are presented within the paper.

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
