# Peer review of "A Comprehensive Review of Small-Molecule Inhibitors Targeting Bruton Tyrosine Kinase: Synthetic Approaches and Clinical Applications"

_molecules, 2023, doi:10.3390/molecules28248037_

Round 1
Reviewer 1 Report
Comments and Suggestions for Authors
The article: A Comprehensive Review of Small-Molecule Inhibitors 2 Targeting Bruton Tyrosine Kinase: Synthetic Approaches and 3 Clinical Applications by Zhao, Wang and collaborators is an important contribution to the important class of new targets for the treatment of auto-immune diseases among others.
The following point should be addressed by the authors.
page 5, Scheme 1 The text describes reaction with malonyl chloride but in the drawing malony-dinitril is shown. It should read Knoevenagel type reaction.
page 6, The text describes phosphorus trichloride but the scheme shows POCl3.
In the section 3.3. Zanubrutinib: sulfoxide chloride should be replaced by thionyl chloride. Malonyl amine should be relaced by Malonyl-dinitril.
page 8: 3.5. Orelabrutinib Please include the off-target toxicities.
page 9: Zanubrutinib: sulfoxide chloride should be replaced by thionyl chloride in the full manuscript.
page 14 3.13. Nemtabrutinib: What is an acetic burst reaction?
Conclusions: What are the major problems with these drugs ?
Comments on the Quality of English Languageminor checks
Author Response
The article: A Comprehensive Review of Small-Molecule Inhibitors 2 Targeting Bruton Tyrosine Kinase: Synthetic Approaches and 3 Clinical Applications by Zhao, Wang and collaborators is an important contribution to the important class of new targets for the treatment of auto-immune diseases among others. The following point should be addressed by the authors.
page 5, Scheme 1 The text describes reaction with malonyl chloride but in the drawing malony-dinitril is shown. It should read Knoevenagel type reaction.
Response: Thanks very much for your suggestion. We have corrected “malonyl chloride” to “malony-dinitril” and changed “Knoevenagel condensation reaction” to “Knoevenagel type reaction”.
page 6, The text describes phosphorus trichloride but the scheme shows POCl3.
Response: Thanks very much for your suggestion. We have corrected “phosphorus trichloride” to “phosphorus oxychloride”.
In the section 3.3. Zanubrutinib: sulfoxide chloride should be replaced by thionyl chloride. Malonyl amine should be replaced by Malonyl-dinitril.
Response: Thanks very much for your suggestion. We have corrected “sulfoxide chloride” to “thionyl chloride” and changed “malonyl” to “malony-dinitril”.
page 8: 3.5. Orelabrutinib Please include the off-target toxicities.
Response: Thanks very much for your suggestion. We have included the toxicities of Orelabrutinib in section 3.5.
page 9: Zanubrutinib: sulfoxide chloride should be replaced by thionyl chloride in the full manuscript.
Response: Thanks very much for your suggestion. We have corrected “sulfoxide chloride” to “thionyl chloride” in the full manuscript.
page 14 3.13. Nemtabrutinib: What is an acetic burst reaction?
Response: Thanks very much for your suggestion. We have carefully reviewed the reaction conditions and rewrote the synthesis method for Nemtabrutinib in section 3.13.
Conclusions: What are the major problems with these drugs?
Response: Thanks very much for your suggestion. We have a section titled "Challenge and Prospective" in the main text, which conclude the major problems with these drugs.
Reviewer 2 Report
Comments and Suggestions for Authors
see attached file

Author Response
The review by Zhang et al deals with small molecules BTK inhibitors and takes into account (as the title says) the clinical applications and the synthesis. The topic may be interesting, but the paper lacks an overall vision: it seems only a separate list of information on 17 molecules. A first point to be clarified is why the authors have chosen only these molecules and not others (see, for instance, doi: 10.3389/fcell.2021.630942). It is necessary to link the various parts: some compounds (which belong to the same structural class) could be discussed together; a table maybe be inserted in place of figure 1, in which the authors report (as a synopsis) for each molecules name, company, potency (IC50) measured on the enzyme, mechanism of action (reversible/irreversible), chemical structure.
Response: Thanks very much for your suggestion. The primary objective of this review is to provide an all-encompassing examination encompassing the clinical application and synthetic routes of 17 representative BTK inhibitors in the clinic. The related topics have never been reported before. Previously published similar reviews were recognized by Eur. J. Med. Chem. and J. Med. Chem (For detailed information, please see J. Med. Chem. 2018, 61, 7004−7031; J. Med. Chem. 2019, 62, 7340−7382; J. Med. Chem. 2022, 65, 9607−9661; Eur. J. Med. Chem. 220 (2021) 113473; Eur. J. Med. Chem. 257 (2023) 115492; Eur. J. Med. Chem. 256 (2023) 115434; Eur. J. Med. Chem. 259 (2023) 115654). According to their chemical structure, we have divided these compounds into the following categories: Aminopyrimidines, Pyrimidine-fused bicyclic heterocycles, Benzopyrroles, Pyrazine-fused bicyclic heterocycles and others. And we have submitted table 1 in the supporting information, which includes the information of each molecules name, company, potency (IC50) measured on the enzyme, mechanism of action (reversible/irreversible).
The paper should be revised by somebody that knows chemistry, to correct the many mistakes. Please consider that the following are only examples and that this list is not exhaustive.
1) In many instances the chemical name of the reactant is wrong: for example, malononitrile is called malonyl chloride (line 36) or malonylamine (line 87); thionyl chloride is called sulfoxide chloride (line 86, line 156).
Response: Thanks very much for your suggestion. Revised.
2) Often the kind of reaction or the role of the reagents are wrong:
the “purification” of Zanu-011 (line 96) is actually an enantiomeric separation.
Response: Thanks very much for your suggestion. We have changed “purified” to “separated”.
“… pyrazine-2-carbonitrile (ACAL-001) is catalytically reduced by Raney Ni …” (line 61): hydrogen is the reducing agent and Raney Nickel the catalyst
Response: Thanks very much for your suggestion. We have changed “Raney Ni” to “hydrogen”.
Line 165: PIRT-07 is not dehydrogenated rather is transformed into the free base
Response: Thanks very much for your suggestion. We have changed “dehydrogenated” to “transformed into the free base”.
“LUXE-006 undergoes a Curtius rearrangement with the intermediate LUXE-007 to obtain Luxeptinib” (lines 184-5) is luxe-007 that undergoes a Curtius rearrangement to give the isocyanate that reacts with amine Luxe-006 to give the urea.
Response: Thanks very much for your suggestion. We have changed “LUXE-006 undergoes a Curtius rearrangement with the intermediate LUXE-007 to obtain Luxeptinib” to “2,4,6-trifluorobenzoyl azide (LUXE-007) undergoes a Curtius rearrangement to give the isocyanate, which reacts with amine LUXE-006 to give the urea Luxeptinib”.
3) The word substitution must be accompanied with a specification (substitution of chloride, line 113, or the 4-Cl of 2,4-dichloro-5-fluoropyrimidine (SPEB-001) is substituted with …).
Response: Thanks very much for your suggestion. We have added “the 4-Cl of” before “2,4-dichloro-5-fluoropyrimidine (SPEB-001) is substituted with …”.
4) Not all the reagents are numbered, but the same reagent can have several codes: for example, acryloyl chloride which is used in the preparation of several molecules, in schemes 3 and 11 it is not numbered, in others it is identified as OREL-009, SPEB-007, ELSU-006, TOLE-014, ABIV-008. This is misleading because the reagent is the same.
Response: Thanks very much for your suggestion. We have numbered “acryloyl chloride” as SPEB-007 and have consistently used this number in the subsequent schemes.
The structure of tolebrutinib is wrong, the synthesis should be corrected
Response: Thanks very much for your suggestion. Revised.
Other remarks:
- Line 37, “.. more effective than chemotherapy”: this should be better explained: the use of BTK inhibitors is also chemotherapy.
Response: Thanks very much for your suggestion. We have deleted “It appears to be safer, more accurate and more effective than chemotherapy”.
- Line 67: the authors cite table S1, where is it?
Response: Thanks very much for your suggestion. We have submitted table S1 in the supporting information.
- Figure 1: this figure (if maintained) can be rearranged to be easily inserted in the paper without need of a separate section.
Response: Thanks very much for your suggestion. We have placed Figure 1 in the supporting information.
- Line 26 (page 4): WM and GvHD have not been defined. Please check the same for all the abbreviation in the paper, including those of chemicals. An abbreviation should be use if it is reported more than once.
Response: Thanks very much for your suggestion. We have provided full names for all abbreviations as required.
- Lines 28-29: does “strong bond” means covalent bond?
Response: Thanks very much for your suggestion. We have changed “strong bond” to “covalent bond”.
- Line 40: the meaning of DMF-DMA is dimethylformamide dimethylacetal.
Response: Thanks very much for your suggestion. Revised.
- Scheme 8, first step: please add the solvent.
Response: Thanks very much for your suggestion. We have added “THF” in scheme 8.
- Please correct ref.33.
Response: Thanks very much for your suggestion. Revised.
- Page 12: it is reported that elsubrutinib has high potency but this value of IC50 is much higher than those reported for other compounds. Is it a direct activity on the enzyme?
Response: Thanks very much for your suggestion. We have deleted “high potency”. Elsubrutinib irreversibly inhibits BTK enzyme activity and blocks BTK-dependent cellular activation.
Reviewer 3 Report
Comments and Suggestions for Authors
The review paper “A Comprehensive Review of Small-Molecule Inhibitors Targeting Bruton Tyrosine Kinase: Synthetic Approaches and Clinical Applications” by Zhang et al. summarized the synthetic scheme and and clinical application of small-molecule BTK inhibitors.
The topic is important, and the manuscript is well written containing the research results and papers so far. Therefore, I believe that the manuscript can be accepted after comments below are addressed.
Comments:
(1)
In addition to the name of the inhibitor, the inclusion of the brand name and code name will improve the quality of this review.
e.g. Ibrutinib (Imbruvica, PCI-32765)
(2)
In the chemical structure of Elsubrutinib in Scheme 10, what is the stereochemistry between indole and piperidine ring?
Also correct the chemical structure in Figure 1.
Author Response
The review paper “A Comprehensive Review of Small-Molecule Inhibitors Targeting Bruton Tyrosine Kinase: Synthetic Approaches and Clinical Applications” by Zhang et al. summarized the synthetic scheme and and clinical application of small-molecule BTK inhibitors. The topic is important, and the manuscript is well written containing the research results and papers so far. Therefore, I believe that the manuscript can be accepted after comments below are addressed.
Comments:
(1) In addition to the name of the inhibitor, the inclusion of the brand name and code name will improve the quality of this review. e.g. Ibrutinib (Imbruvica, PCI-32765)
Response: Thanks very much for your suggestion. We have added the brand names and code names of the drugs as required.
(2) In the chemical structure of Elsubrutinib in Scheme 10, what is the stereochemistry between indole and piperidine ring? Also correct the chemical structure in Figure 1.
Response: Thanks very much for your suggestion. Revised.
Reviewer 4 Report
Comments and Suggestions for Authors
The authors review a collection of BTK inhibitors with the intend to present clinical as well as synthetic data. While I cannot fully judge the chemistry part, the clinical part appears rather superficial and lacks detail.
In the introductory part, there is an adequate general overview of the BTK function and the clinical indications of BTK inhibitors.
The single subchapters are of varying quality. While some subchapters present some specific information, many subchapters contain commonplaces and fail to include more specific information: e.g. p7 l109-110; p12 l232-235; p15 l314-315
In addition, some statements are not correct or overstated:
p11 l189-190: [46] cites a phase II study, not only phase I
p13 l269-270 is clearly overstated with respect to the conclusion of [59] “While fenebrutinib had an acceptable safety profile, the primary end point, SRI-4 response, was not met despite evidence of strong pathway inhibition.”
Some references are missing:
A reference for p13 l271-274 is missing.
A reference for p11 l214-218 is missing.
A citation for accelerated FDA approval of Tolebrutinib is missing: p16 l363-365
Minor shortcomings:
P5 l58-59: FGR stands for “feline Gardner-Rasheed sarcoma viral oncogene homolog” and not for “fetal growth restriction kinase“
P5 l57 ITK stands for Tyrosine-protein kinase and not for “tyrosine kinase”
There are some abbreviations used for which the full name is not given: WM, MZBL, MCL, SLE
Comments on the Quality of English Language
The quality of the English language is mostly fine.
Author Response
The authors review a collection of BTK inhibitors with the intend to present clinical as well as synthetic data. While I cannot fully judge the chemistry part, the clinical part appears rather superficial and lacks detail. In the introductory part, there is an adequate general overview of the BTK function and the clinical indications of BTK inhibitors. The single subchapters are of varying quality. While some subchapters present some specific information, many subchapters contain commonplaces and fail to include more specific information: e.g. p7 l109-110; p12 l232-235; p15 l314-315
In addition, some statements are not correct or overstated:
Response: Thanks very much for your suggestion. We have added or removed some content as required.
p11 l189-190: [46] cites a phase II study, not only phase I
Response: Thanks very much for your suggestion. Revised.
p13 l269-270 is clearly overstated with respect to the conclusion of [59] “While fenebrutinib had an acceptable safety profile, the primary end point, SRI-4 response, was not met despite evidence of strong pathway inhibition.”
Response: Thanks very much for your suggestion. We have included “While fenebrutinib had an acceptable safety profile, the primary end point, SRI-4 response, was not met despite evidence of strong pathway inhibition” in section 3.4.2 and cited the corresponding reference.
Some references are missing:
A reference for p13 l271-274 is missing.
Response: Thanks very much for your suggestion. We have added the reference “Castillo JJ, Treon SP. What is new in the treatment of Waldenstrom macroglobulinemia?, Leukemia. 2019;33(11):2555-2562”.
A reference for p11 l214-218 is missing.
Response: Thanks very much for your suggestion. We have added the reference “C. McDonald, C. Xanthopoulos, E. Kostareli, The role of Bruton's tyrosine kinase in the immune system and disease, Immunology 164 (2021) 722-736”.
A citation for accelerated FDA approval of Tolebrutinib is missing: p16 l363-365
Response: Thanks very much for your suggestion. We have added the reference “R. Orlandi, A.J.D.o.t.F. Gajofatto, Tolebrutinib. Bruton tyrosine kinase (BTK) inhibitor treatment of multiple sclerosis, Drugs Future 47 (2022) 325-336”.
Minor shortcomings:
P5 l58-59: FGR stands for “feline Gardner-Rasheed sarcoma viral oncogene homolog” and not for “fetal growth restriction kinase“
Response: Thanks very much for your suggestion. Revised.
P5 l57 ITK stands for Tyrosine-protein kinase and not for “tyrosine kinase”
Response: Thanks very much for your suggestion. Revised.
There are some abbreviations used for which the full name is not given: WM, MZBL, MCL, SLE
Response: Thanks very much for your suggestion. We have provided full names for all abbreviations as required.
Round 2
Reviewer 2 Report
Comments and Suggestions for Authors
see attached file

Author Response
The paper by Zhang et al has been revised and substantially improved, however it still needs revision.In my previous comments I wrote that the list of mistakes was not exhaustive, and the authors should revise the manuscript throughout. This has not been done. Another list of points to be changed follows.
Lines 211-213: “Tirabrutinib combines important fragments of Ibrutinib and Acalabrutinib…”. The authors should specify which fragments are they talking about. However, tirabrutinib has the same Michael acceptor of acalabrutinib and of branebrutinib, and the same unsubstituted diaryl ether of ibrutinib, zanubrutinib, orelabrutinib and evobrutinib: why limiting the comparison only to 2 molecules? Maybe the sentence is not clear and should be rephrased.
Response: Thanks very much for your suggestion. We have removed the inappropriate description as required.
Scheme 12: this scheme has been modified with respect to the previous version; now the final compound is drawn as enantiomer. However catalytic hydrogenation in the reported conditions can only give a racemic compound. The synthesis must be completed.
Response: Thanks very much for your suggestion. Revised.
Same structures have different names. This happens with BRU-001 and Zanu-002; Zanu-004 and IBRU-002; Bran-005, Tira-11 and Acal-011; OREL-003 and EVOB-004. Malonodinitrile is numbered in scheme 16 and not in scheme 5.
Response: Thanks very much for your suggestion. We have changed the number ZANU-002 to IBRU-001, ZANU-004 to IBRU-002, ACAL-011 and BRAN-005 to TIRA-011, OREL-003 to EVOB-004. We have deleted the number of malony-dinitril in Scheme 16.
Lines 151 and 156: please explain what is Smopex-301 and T3P
Response: Thanks very much for your suggestion. T3P is the abbreviation of ropylphosphonic anhydride. Smopex-301 is a convenient polymer supported version of triphenylphosphine. We have explained what is Smopex-301 and T3P in the manuscript as required.
Lines 201-202: SPEB-007 is ZANU-012 (but ZANU-011 has been skipped)
Response: Thanks very much for your suggestion. We have changed the number SPEB-007 to acryloyl chloride and renumbered Scheme 5.
Line 218: please remove “substitution”.
Response: Thanks very much for your suggestion. Revised.
Line 220: please replace “substitution” with “reaction”.
Response: Thanks very much for your suggestion. Revised.
Line 222: "amidated" is a superficial description of this step, please explain it better.
Response: Thanks very much for your suggestion. Revised.
Line 229: “acidification” should be replaced with “treatment of TIRA-012 with”, or with “salification”
Response: Thanks very much for your suggestion. Revised.
Line 234: please remove the comma after Nebrutinib
Response: Thanks very much for your suggestion. Sorry, we can not find “comma after Nebrutinib”.
Line 243: please replace “was” with “is”.
Response: Thanks very much for your suggestion. Revised.
Line 245: please insert “to” after “added”
Response: Thanks very much for your suggestion. Revised.
Lines 273-4: “Abivertinib, developed by Zhejiang ACEA Pharmaceutical Co., Ltd., has applied to list to treat …” Is this sentence correct?
Response: Thanks very much for your suggestion. Revised.
Line 308: please replace “by” with “using”
Response: Thanks very much for your suggestion. Revised.
Line 377: please insert “acetonitrile using” after “in”
Response: Thanks very much for your suggestion. Revised.
Lines 478 and 480: please replace “substituted” with “reacted”
Response: Thanks very much for your suggestion. Revised.
Line 485: the word "piperidinium" means that the piperidine nitrogen has a positive charge which is not shown in scheme 17. Using “piperidine” would be more appropriate.
Response: Thanks very much for your suggestion. Revised.
Ref 75 should be WO2015048662A2 (A3 is he search report).
Response: Thanks very much for your suggestion. Revised.
The authors have added a table reporting several data; it is a pity, on my opinion, that this table is in the supplementary data and not in the text, because it would help the reader with important information. Nevertheless, it supplementary data are available, it should be stated at the end of the manuscript.
Response: Thanks very much for your suggestion. We have revised them as required by adding a statement at the end of the manuscript.
Reviewer 4 Report
Comments and Suggestions for Authors
The authors sufficiently addressed my comments and revised the manuscript accordingly.
Author Response
Thank you very much for your recognization of our work.